# Universal machine learning for the response of atomistic systems to external fields

Yaolong Zhang[1,3] & Bin Jiang [1,2] ✉

Machine learned interatomic interaction potentials have enabled efficient and accurate molecular simulations of closed systems. However, external fields, which can greatly change the chemical structure and/or reactivity, have been seldom included in current machine learning models. This work proposes a universal field-induced recursively embedded atom neural network (FIREANN) model, which integrates a pseudo field vector-dependent feature into atomic descriptors to represent system-field interactions with rigorous rotational equivariance. This "all-in-one" approach correlates various response properties like dipole moment and polarizability with the field-dependent potential energy in a single model, very suitable for spectroscopic and dynamics simulations in molecular and periodic systems in the presence of electric fields. Especially for periodic systems, we find that FIREANN can overcome the intrinsic multiple-value issue of the polarization by training atomic forces only. These results validate the universality and capability of the FIREANN method for efficient first-principles modeling of complicated systems in strong external fields.

The interplay between external fields and chemical systems is of fundamental importance in a range of physical, chemical, and biological processes[1,2]. By interacting with atoms, molecules, or condensed matter, external (mainly electric) fields can induce electronic/spin polarization and spatial orientation of the system, which have offered a particular means to alter chemical structures[3], promote electron transfer[4], control phase transitions of materials[5] or conformational transformations of biomolecules[6], subtly manipulate chemical reactivity and selectivity in catalysis[7-10] and quantum dynamics in cold chemical reactions[11-13].

Exact field-dependent quantum scattering calculations were feasible only for very small systems[14]. Density functional theory (DFT) and ab initio molecular dynamics (AIMD) simulations based on the modern theory of polarization[15] have been more commonly applied to study more complex aperiodic and periodic systems in the presence of external electric fields[16-20]. However, the AIMD approach remains very demanding, especially when nuclear quantum effects (NQEs) are

important[19]. Although empirical force fields can be instead highly efficient[21,22], their accuracy is limited by empirical functions and approximate expressions for the interaction Hamiltonian. For example, the commonly used dipole-field approximation truncates the perturbation of the system by an electric field to the first order (i.e. only the interaction with the permanent dipole is included) and omits higher-order interactions associated with polarizability, hyperpolarizability, and so on. Moreover, except these reactive force fields[23-25], most of them fall short of describing bond breakage/formation.

Recent years have witnessed revolutionary successes of machine learning (ML) methods in solving high-dimensional problems in chemistry[26-32]. Various ML models for accurately representing potential energy surfaces (PESs) have been developed[33-48]. Some of them have been extended to learn tensorial properties such as the dipole moment and polarizability tensor with correct rotational equivariance[48-60], enabling efficient field-free simulations of electronic and vibrational spectra. However, most ML models treat the potential

[1]Key Laboratory of Precision and Intelligent Chemistry, Department of Chemical Physics, Key Laboratory of Surface and Interface Chemistry and Energy Catalysis of Anhui Higher Education Institutes, University of Science and Technology of China, Hefei, Anhui 230026, China. [2]Hefei National Laboratory, University of Science and Technology of China, Hefei 230088, China. [3]Present address: École Polytechnique Fédérale de Lausanne, 1015 Lausanne, Switzerland. ✉e-mail: bjiangch@ustc.edu.cn

energy and its response properties to electric fields separately, without capturing the field dependence. The influence of electric fields was first taken into account by Christensen et al. in a kernel-based regression method by constructing a fictitious dipole arising from fictitious partial charges and coupling it with the field vector to yield a scalar dipole-field interaction analog[61]. Müller and coworkers incorporated the dipole-field interaction similarly in their FieldSchNet neural network (NN) model to describe the solvent effect in the form of an effective field[62]. Gao and Remsing separated the long- and short-range interactions by introducing self-consistent effective electric fields, which are used as input for subsequent NNs along with local atomic coordinates to describe perturbations to the short-range system from the effective electric field. This strategy however does not rigorously fulfill the rotational equivariance with respect to the field direction[63].

In this work, by introducing a simple field-dependent feature into the description of the atomic environment, we develop a field-induced recursively embedded atom neural network (FIREANN) model with correct rotational equivariance of a system interacting with an external field. Without any truncation of field-induced interactions, FIREANN describes not only the energy variation with applied field strength and direction, but also the associated response properties simultaneously up to (in principle) any order. Path-integral-based molecular dynamics (MD) simulations with well-trained FIREANN models yield reliable ab initio spectroscopy of representative molecular and condensed phase systems in the presence of electric fields. A remarkable characteristic of this model is that it can detour the multivalued issue of polarization in periodic systems, a well-known but largely neglected fact in existing ML models, by learning atomic forces only.

## Results
### FIREANN framework
The FIREANN model is built on top of a physics-inspired recursively embedded atom neural network (REANN) framework[64] that uses embedded atom densities (EADs) as descriptors to atomic environment (Detailed in the Methods section). In the absence of electric fields, EADs are constructed in a quantum chemical spirit by the linear combination of Gaussian-type orbitals (GTOs) of surrounding atoms, preserving the overall rotational, translational, and permutational

invariance of the system. However, an applied field can certainly redistribute the electron density and break down the rotational invariance of the system. The corresponding field-system interaction depends on the direction and strength of the electric field. To characterize this influence in a physically meaningful way, for an applied field ($\vec{\varepsilon}$), we include a virtual field vector-dependent function, namely,

$$\varphi_{l_x l_y l_z}(\vec{\varepsilon}) = (\varepsilon_x)^{l_x}(\varepsilon_y)^{l_y}(\varepsilon_z)^{l_z}. \tag{1}$$

Following the procedure of feature construction in Methods, the field-dependent orbital was combined into the GTO to form a field-induced EAD (FI-EAD) vector that comprises various density values determined by different sets of contracted coefficients,

$$\rho_i^n = \sum_{l=0}^{L} \sum_{\substack{l_x,l_y,l_z \\ l_x+l_y+l_z=l}} \frac{l!}{l_x!l_y!l_z!} \left[ \sum_{m=1}^{N_\varphi} d_m^n \left( \sum_{j\neq i}^{N_c} c_j \varphi_{l_x l_y l_z}^m \left(\vec{r}_{ij}\right) + c_\varepsilon \varphi_{l_x l_y l_z}\left(\vec{\varepsilon}_i\right) \right) \right]^2. \tag{2}$$

Here, the applied field felt by each atom is represented by a position vector of a pseudo-atom relative to that atom ($\vec{\varepsilon}_i$, as illustrated in Fig. 1). The FI-EAD feature can be rewritten in terms of interatomic distances and enclosed angles[39],

$$
\begin{aligned}
\rho_i^n = & \sum_{l=0}^{L} \sum_{j,k\neq i}^{N_c} c_j c_k \left[r_{ij}r_{ik}\right]^l \cos^l(\theta_{ijk}) \sum_{m,m'=1}^{N_\varphi} d_m^n d_{m'}^n f_m(r_{ij}) f_{m'}(r_{ik}) V \\
& + \sum_{l=0}^{L} \sum_{j\neq i}^{N_c} c_j c_e \left[r_{ij}|\vec{\varepsilon}_i|\right]^l \cos^l(\theta_{ij\varepsilon}) \sum_{m=1}^{N_\varphi} d_m^n f_m(r_{ij}),
\end{aligned}
\tag{3}
$$

where we have combined the radial Gaussian and switching functions in $f_m$ for simplicity. From Eq. (3), one immediately realizes that the FI-EAD feature depends not only on atomic coordinates, but also on the field strength ($|\vec{\varepsilon}_i|$) and the closed angle $\theta_{ij\varepsilon}$ between $\vec{\varepsilon}_i$ and each $\vec{r}_{ij}$ vector. In practice, rotating the field or the system separately will lead to different FI-EAD values, while the synchronous rotation of the field and the system without altering the relative direction of $\vec{\varepsilon}$ with respect to each coordinate $\hat{r}_{ij}$ will not. This FI-EAD feature captures the

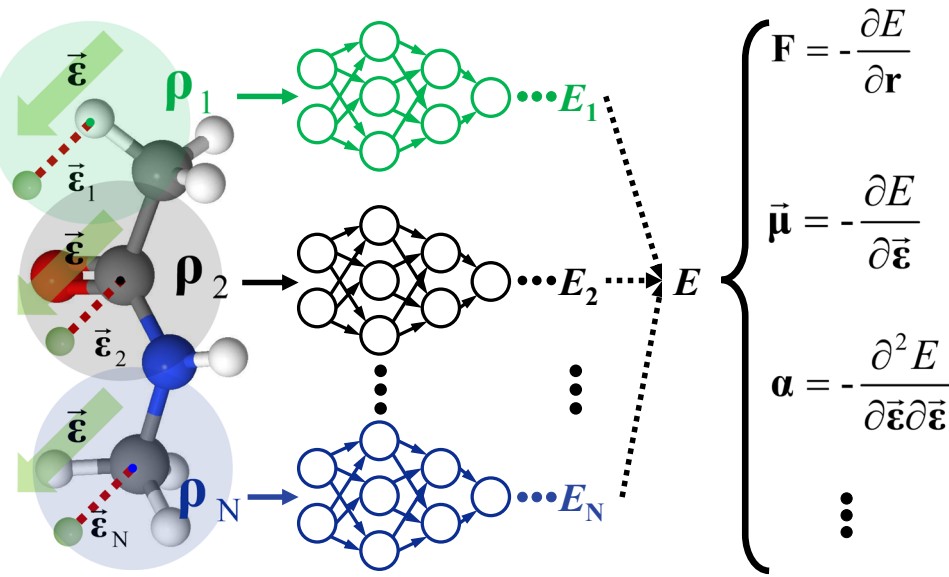

**Fig. 1 | Schematic of FIREANN framework.** The field-induced recursively embedded atom neural network (FIREANN) framework introduces a pseudo atomic field vector ($\vec{\varepsilon}$) relative to each atom (represented by the green transparent atom). These pseudo atomic field vectors mimic the behavior of real atoms and are combined with actual atoms to produce a field-dependent embedded atomic density ($\rho$), which is used as the input of the neural network and yield the field-dependent energy ($E$). Physical quantities like atomic forces (**F**), dipole moment ($\mu$) and polarizability ($\alpha$), correspond to the energy derivatives with respect to coordinates (**r**) or electric field ($\vec{\varepsilon}$). Note that the shaded region represents the local environment of each central atom, defined by a cutoff radius.

nature of the interaction between the system and the applied electric field without changing the physical form of the EAD feature. The resulting rotational equivariance is conserved in any subsequent message passing of the environment- and field-dependent orbital coefficients ($c_j$ and $c_\varepsilon$) and thus in the final potential energy. When $\vec{\varepsilon} = 0$, this model naturally reduces to the original REANN model. The only extra cost for evaluating FIREANN compared to the standard REANN is that of a field-induced orbital, which is almost negligible as evident from Eqs. (1) and (2).

As an extra benefit, the FIREANN framework intrinsically describes the response of the potential energy to an external field up to an arbitrary order by taking the analytical gradients of the potential energy with respect to the field vector. For example, the electric dipole moment ($\vec{\mu}$) is the first and the polarizability tensor ($\boldsymbol{\alpha}$) the second-order response to electric fields,

$$\vec{\mu} = -\frac{\partial E}{\partial \vec{\varepsilon}} \; ; \; \boldsymbol{\alpha} = \frac{\partial \vec{\mu}}{\partial \vec{\varepsilon}} = -\frac{\partial^2 E}{\partial \vec{\varepsilon} \, \partial \vec{\varepsilon}}. \tag{4}$$

These properties can be simultaneously learned in a FIREANN model by adopting the following loss function,

$$L(\mathbf{w}) = \sum_{m=1}^{N_b} \left[ \lambda_V \times \left( E_m^{NN} - E_m^{Ref} \right)^2 + \lambda_F \times \left| \left( -\frac{\partial E}{\partial \mathbf{r}} \right)_m^{NN} - \mathbf{F}_m^{Ref} \right|^2 \right.$$
$$\left. + \lambda_\mu \times \left| \left( -\frac{\partial E}{\partial \vec{\varepsilon}} \right)_m^{NN} - \vec{\mu}_m^{Ref} \right|^2 + \lambda_\alpha \times \left| \left( -\frac{\partial^2 E}{\partial \vec{\varepsilon} \, \partial \vec{\varepsilon}} \right)_m^{NN} - \boldsymbol{\alpha}_m^{Ref} \right|^2 \right] / N_b, \tag{5}$$

where $N_b$ is the size of the batch dataset, the superscripts NN and Ref refer to the NN-predicted and reference quantities, and $\lambda_V$, $\lambda_F$, $\lambda_\mu$, and $\lambda_\alpha$ represent the weights of the energy, force, dipole moment, and polarizability, respectively, in the loss function. Note that these response properties by construction offer correlated information on the field-dependence of the PES rather than being simply accumulated in the loss function[40]. As a result, they have a similar effect as that of forces and hessians, which can help improve the fitting quality. We also note that FIREANN not only applies to electric fields as demonstrated by numerical examples below, but also equally to magnetic fields in the same spirit, by which the magnetic dipole and/or magnetic polarizability can be obtained.

## A toy system
We first take the $H_2O$ molecule as a toy system to verify the symmetry adaption of the FIREANN method subject to an external electric field. A FIREANN model was constructed with just a single equilibrium geometry lying in the $yz$ plane and the electric field being $0.1\,\mathrm{V\,Å^{-1}}$ along the $x$ direction. As displayed in Fig. 2a, when the molecule rotates about the $x$ axis, its potential energy does not change at all as the field is always orthogonal to the molecular plane. This is exactly encoded in FIREANN. On the other hand, the potential energy varies with the molecular rotation about the $y$ axis, as shown in Fig. 2b. The energy variation representing the interaction between the molecular dipole and the electric field is again well predicted by the FIREANN model. Importantly, FIREANN further exhibits excellent extrapolatability in Fig. 2c, where the field intensity along the $x$ axis varies from −0.2 to $0.2\,\mathrm{V\,Å^{-1}}$, resulting in a symmetrical energy dependence on the field direction. The FIREANN model trained with a single data successfully reproduces the energy profile generated by DFT. In comparison, FieldSchNet fails to predict the correct field-induced energy dependence in the same condition and give a constant energy as shown in Fig. 2c. More detailed comparisons between FIREANN and FieldSchNet will be discussed below.

## Molecular spectroscopy
A distinct feature of the proposed FIREANN model is its all-in-one predictions for energies (atomic forces) and response properties with and without an electric field. We first demonstrate this feature for the $N$-methylacetamide (NMA) molecule, which has been widely used as a model system of the amide group to construct spectroscopic maps and simulate the spectra of the peptide backbone[52,65–68]. Specifically, we constructed an FIREANN model by learning a mix set of ab initio energies, forces, dipole moments, and polarizabilities for the NMA molecule in an electric field varying from 0.0 to $0.4\,\mathrm{Å^{-1}}$ along $x$ direction. Figure 3 clearly shows that the universal FIREANN model achieves an excellent accuracy for energy, dipole moment, and polarizability, with corresponding root mean square errors (RMSEs) of 0.0053 eV, 0.028 Debye, and 0.51 a.u., respectively. Given the synchronous prediction of these quantities, the FIREANN model enables efficient MD simulations of IR and Raman spectra in comparison with experimental data.

Figure 4a compares the calculated and experimental field-free infrared (IR) spectra[69] for NMA at 300 K. In general, the classical MD-based result agrees reasonably well with the experimental spectrum, even reproducing double peaks for the C-O stretching vibration (~1710 cm⁻¹) corresponding to the well-known P/R rotational structure-induced splitting[69]. However, the calculated bands of Amide II, Amide III, Amide A (the N-H stretch band, ~3507 cm⁻¹), and the band including C-H stretching mode and other bending overtones of the methyl group (~2950 cm⁻¹) are apparently blue-shifted compared to experiment. This discrepancy is likely due to the neglect of NQEs in the classical treatment of these vibrational bands relevant to hydrogen atoms. To solve this problem, path-integral based thermostated ring polymer

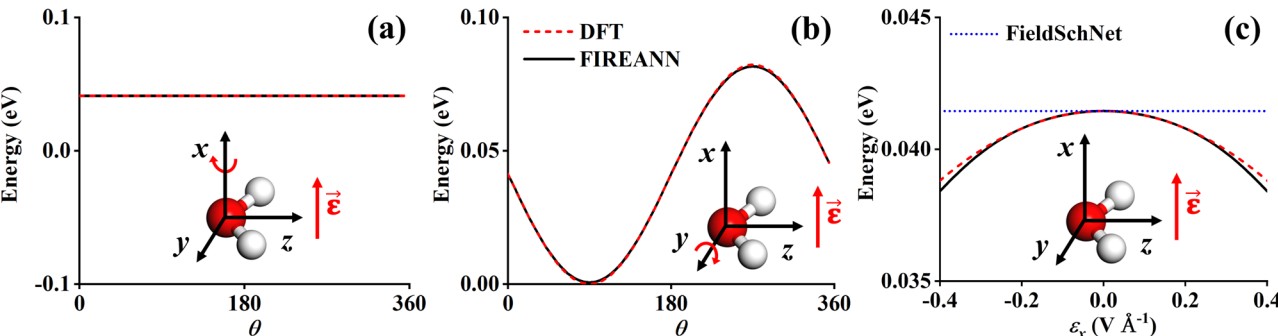

**Fig. 2 | Rotational and field intensity dependence of the FIREANN model.** Comparison of energy curves of a water molecule lying on the $yz$ plane rotating about the **a** $x$ axis and **b** $y$ axis calculated by density functional theory (DFT) and field-induced recursively embedded atom neural network (FIREANN), where an electric field ($\vec{\varepsilon}$) with the intensity of $0.1\,\mathrm{V\,Å^{-1}}$ is applied along the $x$ axis; **c** DFT and FIREANN energy curves varying with the electric field intensity ($\varepsilon_x$), compared with FieldSchNet[62]. Source data are provided as a Source Data file.

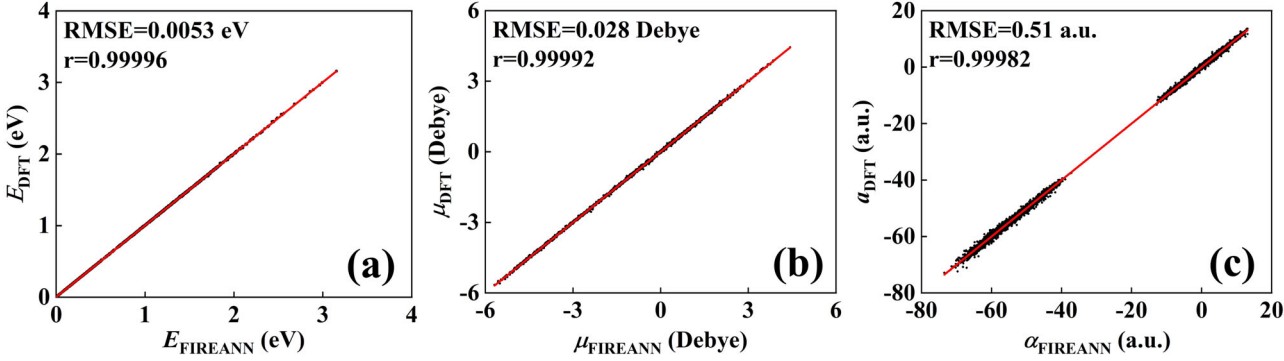

**Fig. 3 | Performance of the FIREANN model for NMA.** Correlation plots of **a** potential energies, **b** dipole moments, and **c** polarizabilities based on field-induced recursively embedded atom (FIREANN) predictions ($E_{FIREANN}$, $\mu_{FIREANN}$ and $\alpha_{FIREANN}$, respectively) and Density functional theory (DFT) data ($E_{DFT}$, $\mu_{DFT}$, and $\alpha_{DFT}$, respectively) in the test set of N-methylacetamide (NMA). The root mean square error (RMSE) and Pearson Correlation Coefficient ($r$) are shown in the corresponding panel. Source data are provided as a Source Data file.

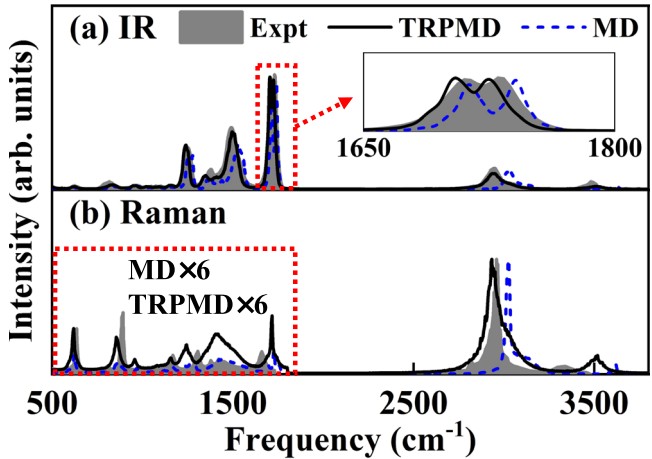

**Fig. 4 | Field-free vibrational spectra of NMA.** Comparison of experimental (Expt), molecular dynamics (MD), and thermostated ring polymer molecular dynamics (TRPMD) based **a** infrared (IR) spectra and **b** Raman spectra of the N-methylacetamide (NMA) molecule at 300 K. **a** The inset zooms in the C-O stretching vibration peak ~1700 cm$^{-1}$ to show the P/R rotation branches more clearly. In panel (b) the intensity of the calculated Raman spectra in the low-frequency region is amplified by a factor of 6 for qualitative comparison with the measured liquid spectrum[71] due to the lack of the experimental spectrum for a single NMA molecule. Source data are provided as a Source Data file.

molecular dynamics (TRPMD)[70] simulations were performed. The TRPMD result significantly improves the agreement with the experiment and reproduces most bands in not only their positions but also their shapes and intensities. Figure 4b compares the calculated and experimental resonance Raman spectra of NMA for zero field. Due to the lack of the experimental spectrum for a single NMA molecule, the measured liquid spectrum[71] is taken form qualitative comparison. Encouragingly, while there is apparent mismatch regarding the relative peak intensities of low-frequency modes, the TRPMD spectrum reproduces most of the observed bands reasonably well.

FIREANN also predicts the in-field molecular spectra, as shown in Fig. 5, where the field strength increases from 0 to 0.4 V Å$^{-1}$ every 0.1 V Å$^{-1}$ along $x$ direction. In these in-field IR spectra, the C-O stretching band seems most influenced by the applied field. As mentioned in the field-free spectrum, this band has an intrinsic P/R rotational double-peak structure. Interestingly, with increasing field strength, this P/R branch splitting gradually vanishes and the absorption peak gets narrower and higher. This phenomenon implies interplay between the electric field and the molecular rotation. Indeed,

the dipole moment of NMA, which is almost parallel to the C-O bond[69,72], tends to reorient to the opposite direction of the electric field to minimize the energy. Increasing the field intensity increases the dipole-field interaction and more strongly confines the NMA molecule in the preferable orientation. In addition, a significant red shift of the CO stretching vibration is found roughly proportional to the field intensity. This redshift is likely a natural consequence of the weakening of the chemical bond by the applied electric field. A similar but smaller redshift is also found for the N-H stretching, consistent with the fact that the electron cloud of the C-O group is more polarizable than the N-H group, due apparently to the higher electron density there. Furthermore, we decompose the molecular polarizability into isotropic ($\alpha_{iso} = \text{tr}(\boldsymbol{\alpha})/3$) and anisotropic ($\boldsymbol{\alpha}_{aniso} = \boldsymbol{\alpha} - \alpha_{iso}\mathbf{I}$) terms and exhibit corresponding Raman spectra in Fig. 5, respectively. The anisotropic Raman spectra show a similar field-dependence of the C-O stretching band for the same reason. However, the rotational splitting is absent in isotropic Raman spectra as the isotropic polarizability is rotation invariant. As a result, the increasing field results in only a pure red shift of the C-O stretching vibration.

## Liquid water

The FIREANN model is by its atom-wise form capable of describing the response of periodic systems to external electric fields. We test this capability in liquid water. However, unlike molecular systems, the polarization (dipole moment per unit volume) of a periodic system is a multivalued quantity according to the modern theory of polarization[15], resulting in multiple parallel branches that differ by a polarization quantum represented by the product of any lattice vector and the electronic charge and divided by the volume of the lattice[15]. This ill-defined multiplicity may lead to sudden jumps in the dipole moment. Figure 6a shows clearly the abrupt discontinuities in the evolution of the $x$ component of DFT calculated dipole moment along an AIMD trajectory without an electric field. This accidental change in the dipole moment poses challenges for conventional atomistic ML models that learn field-free dipoles, which typically decompose the global dipole moment vector into local atomic dipoles and represent atomic dipoles by the product of atomic charges and position vectors[49,50,52]. Importantly, this discontinuity issue occurs more frequently under a high field strength, as seen in Fig. 6b. Likewise, the in-field total energy is supposed to be discontinuous at these configurations as the dipole-field interaction jumps. Schienbein also recognized the multiple-valued problem of the dipole moment and proposed to learn the atomic polar tensor which is the spatial derivative of dipole instead of learning the dipole itself. These smooth spatial derivatives can be transformed into time derivatives of dipole in MD to calculate autocorrelation functions, ultimately yielding the IR spectrum[60]. Similarly,

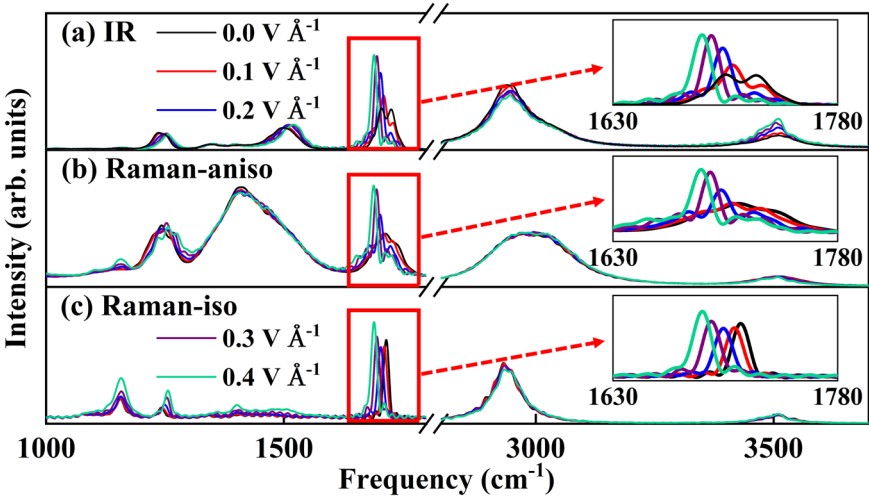

**Fig. 5 | In-field vibrational spectra of NMA predicted by FIREANN.** Comparison of thermostated ring polymer molecular dynamics based **a** infrared (IR) spectra, **b** Raman anisotropic (Raman-aniso), and **c** Raman isotropic (Raman-iso) spectra of N-methylacetamide at 300 K varying with the external electric field intensity. The inset in each panel zooms in the corresponding C-O stretching vibration peaks. Note that the high-frequency bands above 2800 cm$^{-1}$ **a**–**c** are multiplied by a factor of 10, 0.1, and 0.05 to show the peaks in a similar scale. Source data are provided as a Source Data file.

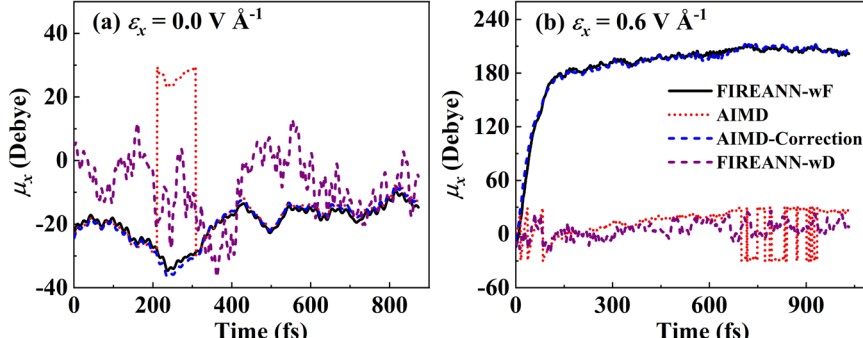

**Fig. 6 | Multi-valued dipole moments in liquid water.** Comparison of $x$ component of dipole moments ($\mu_x$) calculated with force-only training (FIREANN-wF) and brute-force dipole moment training (FIREANN-wD) based on the field-induced recursively embedded atom neural network (FIREANN) with density functional theory (DFT) and corrected DFT data (AIMD-correction) along the ab initio molecular dynamics (AIMD) trajectory and as a function of time, for **a** zero field and **b** an electric field along $x$ axis ($\varepsilon_x$) with the intensity of 0.6 V Å$^{-1}$. Note that the FIREANN-wF predicted dipole moment, which is deduced by the energy gradient to the electric field vector introduces an undetermined constant factor, is shifted by a constant to match the DFT value. Source data are provided as a Source Data file.

learning the offsets of the Wannier centers relative to the corresponding O atoms can overcome this problem[73], however, localizing Wannier centers themselves in more complex systems with strong non-local features, is not a trial task, which easily gets stuck in local minima and has severe convergence difficulties[74]. Furthermore, unlike FIREANN, these previous ML models are designed for field-free systems only, which does not describe the general response of a system to external fields and the field-dependent potential energy surface.

In the FIREANN framework, alternatively, this issue can be easily bypassed by training atomic forces only in the presence of electric field, because the gradient of the energy is actually unaffected. Although the dipole moment and polarizability are not explicitly involved, the interaction between the electric field and the system can be learned implicitly in the force-only training. The dipole moment can then be retrieved by the first-order gradient of the energy output with respect to the field vector. It should be noted that training atomic forces only will introduce an undetermined field-dependent constant to the total energy in our model. This will lead to certain undetermined constants to absolute thermodynamic properties. However, atomic forces (or any properties' gradients with respect to atomic coordinates) are well represented by our model so that the changes of

thermodynamic properties under a given electric field can still be correctly described, which are physically more meaningful in practical calculations. In addition, by including the polarizability in the loss function during the training process, one can eliminate the field dependence of the undetermined constant on the total energy. This allows us to compare energies and the changes in thermodynamic properties under different field strengths, as will be discussed later in the comparison with FieldSchNet.

To validate this strategy, we have constructed a FIREANN model of bulk water including 64 water molecules under an electric field up to 0.6 V Å$^{-1}$ along $x$ direction, using atomic forces as targets only (named FIREANN-wF hereafter). Our model yields accurate predictions for atomic forces, with an overall RMSE of 39.4 meV Å$^{-1}$. In addition, the TRPMD-calculated field-free radial distribution functions (RDFs) of liquid water agree very well with previous on-the-fly results at the same DFT level[75] and experimental data[76], as shown in Fig. 7, further validating the accuracy of the FIREANN-wF potential. It is also beneficial to compare the dipole moments predicted by FIREANN-wF with DFT data. Since dipole moments should smoothly change as the configuration evolves, it is reasonable to correct any abrupt changes in the dipole moment calculated by DFT along an AIMD trajectory. This correction

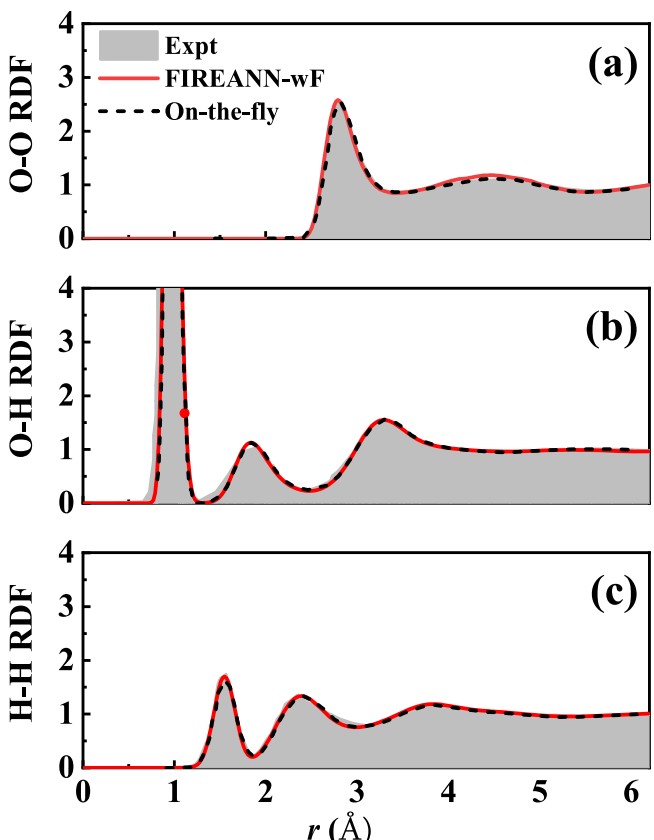

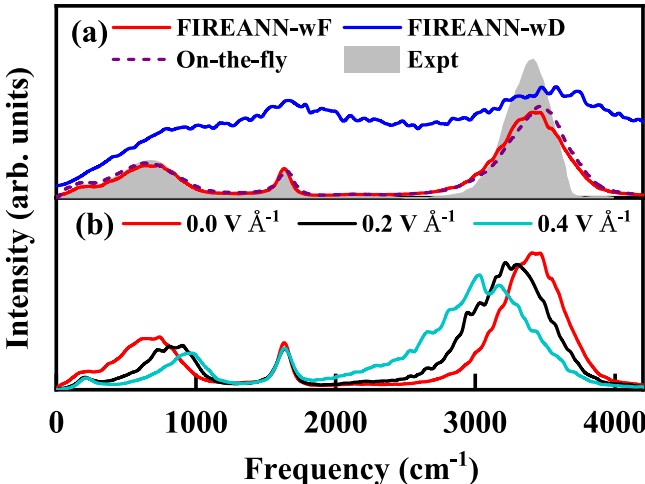

**Fig. 8 | Field-free and in-field IR spectra of liquid water. a** Comparison of experimental[77] (Expt) and three thermostated ring polymer molecular dynamics (TRPMD) based infrared (IR) spectra of liquid water in the absence of electric field at 300 K, computed with force-only training (FIREANN-wF) and brute-force dipole training (FIREANN-wD) based on the field-induced recursively embedded atom neural network model, and on-the-fly calculations at the same computational level extracted from ref. 75 (On-the-fly). **b** Comparison of TRPMD-based IR spectra of liquid water obtained from the FIREANN-wF model with electric fields of 0.0 V Å⁻¹, 0.2 V Å⁻¹, and 0.4 V Å⁻¹ along the *x* direction. Source data are provided as a Source Data file.

**Fig. 7 | Radial distribution functions of liquid water.** Comparison of the experimental[76] (Expt) and theoretical **a** O-O, **b** O-H and **c** H-H radial distribution functions (RDFs) for liquid water at 300 K. Theoretical results are based on thermostated ring polymer molecular dynamics simulations with the force-only training field-induced recursively embedded atom neural network (FIREANN-wF) models or with on-the-fly force calculations at the same computational level extracted from ref. 75 (On-the-fly). Source data are provided as a Source Data file.

involves shifting the dipole moment by an integer multiplied by the product of the corresponding lattice vector and electronic charge. By applying this correction, one ensures that the adjusted dipole moment remains closest to its value in the previous step, allowing for a continuous variation of dipole moments along the trajectory. Impressively, as shown in Fig. 6, the corrected DFT dipole moments perfectly match the FIREANN-wF predictions, without any prior knowledge of these positions of sudden jumps. Interestingly, the FIREANN-wF model captures the drastic increase in the total polarization of the system under an intensive electric field, as shown in Fig. 6b. This is because of additivity of the molecular dipole moment as each water molecule tends to align its dipole moment to the field vector. Figure 6 also shows the poor performance of a flawed model training with multi-branched dipole moments in a brute-force way (named FIREANN-wD hereafter), which obviously fails to follow the correct evolution of the dipole moment. Note that dipole moments in both two models are obtained by calculating the energy gradient with respect to the electric field. It is worth stating that although it is viable to correct the dipole moment along a single AIMD trajectory because the configuration change is minor in adjacent steps, it is difficult to do so in practice for uncorrelated trajectories or for independent single-point calculations. In latter cases, the training dataset will likely include dipole moments of unpredictable branches and give large noises to conventional ML dipole models relying on the multiplication of atomic charges and position vectors.

Misrepresenting the dipole moment surface could have a significant influence on the resultant IR spectrum. Figure 8a compares the

calculated and experimental IR spectra of liquid water at room temperature without an electric field. Thanks to the inclusion of NQEs by TRPMD, the FIREANN-wF model that offers both the correct PES and dipole moment surface, does capture well all experimental vibrational features[77] including the O-H stretching (~3600 cm⁻¹), H-O-H bending (~1690 cm⁻¹), librational (~700 cm⁻¹)[78] and H-bonding stretching bands (~170 cm⁻¹)[78]. Our results also agree well with previous theoretical ones obtained by on-the-fly TRPMD at the same DFT level[75], likely with their DFT dipole moments corrected. By contrast, the IR spectrum predicted by the FIREANN-wF model using the same trajectories deviates significantly from the experimental counterpart. This comparison clearly highlights the necessity of using an ML model fulfilling the physical requirement of the dipole moment in predicting IR spectra. Finally, we show in Fig. 8b the predictions of the FIREANN-wF model for the IR spectrum with an electric field up to 0.4 V Å⁻¹ along *x* direction. Interestingly, the electric field influences mostly the O-H stretching band, resulting in a progressive red shift upon with the increasing field intensity. Unlike the NMA molecule, the red shift here is not solely because of the softening of the O-H bond by the electric field, but also the more ordered structure as a result of the field-induced reorientation of water molecules to be parallel to the field vector[20]. This effect will render the liquid water structure more ice-like[20], in which the O-H vibrational band is lower in frequency. In contrast, this ordering effect hinders the librational rotation of water molecules and naturally results in an increased frequency of the librational mode. In comparison, the H-O-H bending mode is barely affected by the electric field, since the bending motion leads to little change in the direction of the dipole moment.

## Comparison with previous models

Although similar force-only training can be done using previous models proposed in refs. 61,62, their ways of incorporating the external field are completely different from the present FIREANN model, rendering the absence of important high-order field-system interactions. Specifically, these models consider the response of the system to the external field by adding the dot product of a virtual

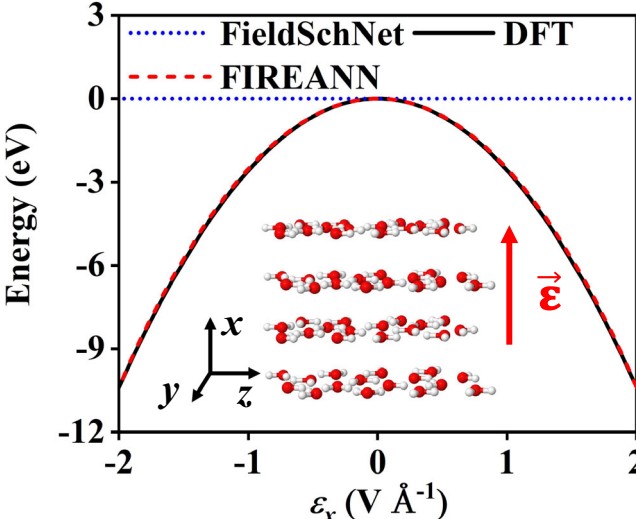

**Fig. 9 | Field-dependent energy curves of FIREANN and FieldSchNet.** Density function theory (DFT) and field-induced recursively embedded atom neural network (FIREANN) energy curves varying with the electric field intensity along $x$ axis ($\varepsilon_x$), compared with FieldSchNet[62] for liquid water. The inset is the employed structure of liquid water. Note that in order to compare the energy curves generated by the different methods, the energy curves are shifted separately so that the field-free energy becomes the zero point of the energy. Source data are provided as a Source Data file.

atomic dipole and the electric field vector (namely $\vec{\mu} \cdot \vec{\varepsilon}$, a scalar value) to the standard atomic descriptor. By construction, high-order field-system interactions are missing in their descriptors. In contrast, by introducing a virtual field-dependent atomic orbital in our field-induced EAD descriptor, we capture the response of the electron density to the external field through orbital-orbital interactions (Eq. (2) in this work). In such a way, all interactions between the field and the atomic environment are included in our FIREANN model. This is a fundamental improvement over all previous models, which can be easily adapted to other equivariant features based ML models[44,48,53] without altering their fundamental architectures. Indeed, Christensen et al.[61] have clearly admitted that their kernel-based model cannot predict polarizability and other high-order response properties. Similar deficiencies arise in the FieldSchNet model as well. Although the nonlinear message passing NNs imposed in that model may learn part of high-order interactions, this incompleteness will cause qualitative failures of FieldSchNet in some cases.

To show this explicitly, we have applied the FieldSchNet package and compared its predictions with present ones using exactly the same dataset. As already presented in Fig. 2c, the FIREANN model perfectly captures the nonlinear energy variation of a water molecule as a function of the field strength. In contrast, the FieldSchNet model predicts no dependence of the energy on the field strength at all. This is because the water molecule lies in the $yz$ plane so all atomic dipoles derived from FieldSchNet always lie in plane, resulting in zero coupling with the electric field applied along the $x$ direction and a constant energy. In practice, all $x$ relevant components in any response quantities (dipole moment, polarizability, etc.) predicted by FieldSchNet are zero.

This phenomenon is not limited to molecules but also applies to periodic systems. To show this, we construct an exemplary dataset consisting of 200 liquid water configurations exposed to an electric field along the $x$ direction ranging from 0 to 0.6 V Å$^{-1}$. Water molecules in these configurations are aligned in four evenly spaced layers (16 molecules in each) perpendicular to the $x$ axis with the first and third layer equal to the second and fourth layers respectively, as shown in

the inset of Fig. 9. To enable the comparison of field-dependent energies, we trained a FIREANN model and a FieldSchNet model respectively with both forces and polarizabilities (to eliminate the field dependence of the undetermined constant of the total energy), and aligned their field-free energies to the same zero point. The difference between FIREANN and FieldSchNet models is amplified in this system, where the RMSEs predicted on the test set by FIREANN and FieldSchNet are 54.5 meV Å$^{-1}$ (2.1 a.u.) and 245.4 meV Å$^{-1}$ (165.1 a.u.) for forces (polarizability), respectively. Again, the FIREANN model precisely captures the large energy variance up to an applied electric field of ±2 V Å$^{-1}$, while the FieldSchNet energy remains constant and deviates from the correct DFT result by several eV, as displayed in Fig. 9. This result also validates the generalizability of the FIREANN model towards representing high-intensity external fields.

We note that this deficiency will generally appear in any configuration if all atomic dipoles along the applied field direction are zero, which would lead to an unphysical behavior near the corresponding configuration space and inevitable large fitting errors. For example, using the same full dataset of liquid water in this work and training with atomic forces and polarizabilities, the FIREANN and FieldSchNet models yield test RMSEs of 45.5 meV Å$^{-1}$ (2.5 a.u.) and 184.7 meV Å$^{-1}$ (12.9 a.u.), respectively. The much worse performance of FieldSchNet represents an indicator of its incomplete description of the field-system interaction. It is worth noting that our FIREANN implementation is more efficient than FieldSchNet, with a training time of 2.4 versus 7.6 minutes per epoch, when running on a single A100 GPU with a memory capacity of 80 GB.

## Discussion

In this work, we have proposed a simple, accurate, and universal FIREANN model to learn the external field-dependent PES and response properties with the proper rotational equivariance. This model allows us to obtain all ingredients from one single training for modeling spectroscopy and dynamics of chemical systems with and without external electric fields. The validity of this model is supported by the good agreement between the predicted vibrational spectra of the NMA molecule and liquid water and field-free experimental data. Moreover, the field-induced alignment of the dipole moment and the softening of the covalent bond are clearly predicted in the in-field IR or Raman spectra. For periodic systems like liquid water, in particular, the intrinsic multi-valued polarization of the system results in the discontinuous dipole moments in the training data and makes it difficult to being represented by conventional machine learning models based on atomic charges. This issue is nicely bypassed in the FIREANN model by learning atomic forces only, which can yield both field-dependent potentials and dipole moments, and thus IR spectra of liquid water. Our results not only clearly validate the high accuracy of the all-in-one FIREANN model, but also elucidate the interplay between chemical systems and electric fields.

In the current implementation built on the original PyTorch[79] framework, training a FIREANN model in the most complete scenario (including energy, forces, dipole, and polarizability tensor) will take 4 times longer than force-only training, as the former process requires sample-to-sample (high-order) gradients. This issue can be largely alleviated by an improved implementation based on the more recently released functorch[80] module in a new version of PyTorch, which allows efficient computation of sample-to-sample (high-order) gradients. Although all results presented in this work are relevant to the system exposed to an electric field, the FIREANN framework can be extended to describe the response of the system to a magnetic field or even to an electromagnetic field by introducing another field vector-dependent virtual function in Eq. (2). This will allow a more complete description of magnetic fields interacting with the system than in ref. 62. Note that the current version of the FIREANN model is limited to describing the influence of a homogeneous external field. In the case of a non-uniform

external field, the response of the electron density to the field is spatially dependent and must be explicitly considered. A feasible way is to discretize the non-uniform field to each atomic center and introduce a nonequivalent field-dependent function to each FI-EAD feature (as implicitly implied in Fig. 1) to approximate the response of each atomic density to the local field experienced by the central atom. Note that this adjustment is intended to introduce an external inhomogeneous field interacting with the entire system. This differs from an inhomogeneous electric field approximately generated by solvent environments, which acts only onto the embedding molecular center as described in ref. 62. These desirable features make the FIREANN approach very promising to efficiently modeling strong field-induced phenomena such as electrochemistry[81,82], plasmonic chemistry[83], and tip-induced catalytic reactions[8].

## Methods

### REANN

The regular REANN model was proposed for representing field-free PESs[64]. Like all atomistic NN models, the total potential energy ($E$) is expressed in the sum of atom-wise contributions, and each atomic energy ($E_i$) is learned by feeding a vector of atomic features for describing the atom-centered environment to an atom-wise NN. In the REANN model, EAD atomic features are specified to include many-body correlations between the central and neighbor atoms, which are simply evaluated by the square of the linear combination of a set of contracted GTOs located at neighbor atoms,

$$\rho_i^n = \sum_{l=0}^{L} \sum_{l_x,l_y,l_z}^{l} \frac{l!}{l_x! l_y! l_z!} \left[ \sum_{j \neq i}^{N_c} c_j \sum_{m=1}^{N_\varphi} d_m^n \varphi_{l_x l_y l_z}^m \left( \vec{\mathbf{r}}_{ij} \right) \right]^2, \quad (6)$$

where the primitive GTO takes the following form,

$$\varphi_{l_x l_y l_z}^m \left( \vec{\mathbf{r}}_{ij} \right) = \left( x_{ij} \right)^{l_x} \left( y_{ij} \right)^{l_y} \left( z_{ij} \right)^{l_z} \exp\left[ -\alpha_m \left( r_{ij} - r_m \right)^2 \right] f_c \left( r_{ij} \right), \quad (7)$$

and the contraction combines different shapes of primitive GTOs together,

$$\chi^n \left( \vec{\mathbf{r}}_{ij} \right) = \sum_{m=1}^{N_\varphi} d_m^n \varphi_{l_x l_y l_z}^m \left( \vec{\mathbf{r}}_{ij} \right), \quad (8)$$

In practical implementation, we reorder the summation over $c_j$ and $d_m^n$ in Eq. (6) to obtain better efficiency. It should be noted that an EAD feature vector ($\boldsymbol{\rho}_i$) consists of a number of density values generated from different sets of contracted GTOs. Although GTOs in our model are expanded in Cartesian coordinates, they can also be expressed in terms of spherical harmonics, resembling in spirit those equivariant features based on spherical harmonics[44,48,53]. Specifically, $\vec{\mathbf{r}}_{ij} = \vec{\mathbf{r}}_i - \vec{\mathbf{r}}_j$ is the position vector (three components) of the central atom $i$ relative to the $j$th neighbor atom with $r_{ij}$ ($x_{ij}, y_{ij}, z_{ij}$) being its norm (Cartesian component), $l = l_x + l_y + l_z$ specifies the orbital angular momentum (e.g., $l = 0$ for the $s$ orbital, $l = 1$ for the $p$ orbital, etc.), $\alpha_m$ and $r_m$ are hyperparameters that determine the center and the width of the radial Gaussian function. In the combination to form an EAD feature, $L$ is the maximum orbital angular momentum of GTOs, $N_\varphi$ is the number of primitive GTOs for each $l$ and $d_m^n$ is the contraction coefficient of the $m$th primitive GTO for the $n$th component of the EAD vector, $N_c$ is the number of neighbor atoms and $c_j$ is the $j$th atomic orbital coefficient within a cutoff radius ($r_c$), $f_c(r_{ij})$ is a cosine-type switching function continuously decaying interatomic interactions to zero at $r_c$ up to second-order derivatives. In particular, realizing that $c_j$ itself necessarily depends on its atomic environment, we express it as the output of an atomic NN based on EAD features centered at atom $j$.

Apparently, REANN is essentially a message-passing NN by recursively expanding the environment-dependent orbital coefficients like this, which has proven an efficient way to incorporate high-order many-body correlations in the local environment[64].

### Training details

All FIREANN models in this study utilize NN with two hidden layers, each containing 64 neurons in each iteration of message-passing. Eight radial functions and $L$ up to 2 were used to construct EAD features with sufficient representability. The initial learning rate was set to 0.002 and decays by a factor of 0.5 whenever the validation loss does not decrease for 100 consecutive epochs. Training stops when the learning rate drops below $1 \times 10^{-5}$. The number of message-passing iterations for $H_2O$, NMA, and liquid water were set to 0, 4, and 3, respectively. The cutoff distances for the three systems were 3.0 Å, 6.0 Å, and 5.0 Å. Other parameters were automatically optimized during the training process. These weights for individual properties in the loss function were dynamically adjusted during the training process. For the NMA molecule, $\lambda_V$, $\lambda_F$, $\lambda_\mu$, and $\lambda_\alpha$ decay linearly from 0.1, 50, 10, and 10 to 0.1, 0.5, 0.5, and 0.5 as the learning rate decays. The same set of weights were used for the $H_2O$ molecule, except that there is no force weight included. As for the liquid water, only atomic forces were trained and weighting was unnecessary.

### Computational details and datasets

Three systems are used to validate the FIREANN model, including a toy system ($H_2O$ monomer), NMA, and liquid water.

### A toy system

The training set of the $H_2O$ molecule contains merely a single equilibrium geometry lying in the $yz$ plane with an electric field of 0.1 V Å$^{-1}$ along the $x$ direction. The potential energy, dipole moment, and polarizability of $H_2O$ were calculated by Gaussian 09[84] at the B3LYP/cc-pVDZ level[85] and used as targets in the loss function defined in Eq. (7).

### NMA

Over 13000 configurations were sampled from canonical ensemble (NVT) classical & path-integral MD simulations at 300 K in the presence of an electric field ranging from 0.0 to 0.4 V Å$^{-1}$ along the $x$ direction and calculated using Gaussian 09[84] at the B3LYP/aug-cc-pVDZ level[85] with D3 correction of dispersion[86]. The dataset was divided into training set, validation set, and test set with a ratio of 8:1:1. Again, the potential energy, atomic force, dipole moment, and polarizability were trained simultaneously.

### Liquid water

A cubic box of 64 water molecules was used in the data sampling. The dataset consists of ~33000 configurations sampled from NVT classical and path-integral AIMD simulations at 300 K with the external electric field ranging from 0.0 to 0.6 V Å$^{-1}$ along $x$ direction. Electronic structures and properties were calculated by CP2K[87] with a hybrid density functional revPBE0[88,89] including D3 correction of dispersion[86]. Goedecker–Tetter–Hutter pseudopotentials[90] with a cutoff of 1200 Ry and a TZV2P basis set were used. Only atomic forces were used to construct the PES, and the dipole moment was excluded due to its discontinuity caused by the multiple-value nature of the periodic systems. To illustrate the deficiency of FieldSchNet, a special dataset was collected consisting of 200 liquid water configurations exposed to an electric field along the $x$ direction ranging from 0 to 0.6 V Å$^{-1}$. Water molecules in these configurations were averagely placed in four evenly spaced layers perpendicular to the $x$ axis (16 molecules in each). In addition, the first (second) layer was made identical to the third (fourth) layer. In this way, the sum of dipole moments in each layer was kept in plane and any interlayer dipole moment canceled out, leaving a zero $x$ component of the total dipole moment. In the data sampling,

water molecules were centered evenly spaced grids (4 × 4) in the plane with some random shifts within 0.3 Å and random in-plane orientation. The intramolecular O-H bond lengths and H-O-H angle of each molecule are randomly displaced from their equilibrium values within ~0.1 Å and ~2°.

## MD simulations with FIREANN models

**NMA.** To compare the calculated IR and Raman spectra of NMA with experimental data, classical MD simulations were performed at 300 K in the absence of an electric field. The NMA molecule was first equilibrated with 20 ps using the Andersen thermostat[91], after which two-hundred snapshots with corresponding momentum were randomly chosen for initializing subsequent NVE MD simulations of 25 ps. The time correlation functions (TCFs) were computed by the average of 200 such trajectories. IR and Raman spectra were obtained by the Fourier transform of the TCFs of the time derivatives of dipole and polarizability, respectively[92]. In addition, to include NQEs, path-integral based TRPMD simulations[70,93] were performed with Langevin thermostats attached to all non-centroid normal modes, with and without adding an electric field. Other computational details are similar to those in the classical MD simulations. The resulting field-dependent IR and Raman spectra were obtained by the Fourier transform of the centroid-based TCFs on an average of 200 TRPMD trajectories. In all simulations, the time step was kept at 0.1 fs.

**Liquid water.** The system consists of 64 $H_2O$ molecules with a side length 12.4185 Å. The NVT classical MD simulations of liquid water were performed at 300 K, using Andersen thermostat[91]. To obtain convergent IR spectra, we extracted 128 positions and momentum from an equilibrium NVT trajectory as initial states for NVE MD simulations, with a total time of 20 ps per NVE simulation and a time step of 0.1 fs. The same setup was used for a 24-bead TRPMD simulations to include NQEs, which was found to converge the spectroscopic results.

## Data availability

The dataset of NMA molecules and the exemplary dataset of liquid water (200 structures) generated in this study have been deposited in the github [https://github.com/zhangylch/FIREANN/tree/main/data]. The initial and final structures of MD/TRPMD simulations generated in this study have been deposited in the github [https://github.com/zhangylch/FIREANN/tree/main/data/md_stru]. The complete liquid water data are available upon reasonable request from the corresponding author. The data generated in this study are provided in the Source Data file. Source data are provided with this paper.

## Code availability

The FIREANN package are available from https://github.com/zhangylch/FIREANN[94].

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

## Acknowledgements

This work is supported by the Strategic Priority Research Program of the Chinese Academy of Sciences (XDB0450101), Innovation Program for Quantum Science and Technology (2021ZD0303301), CAS Project for Young Scientists in Basic Research (YSBR-005), National Natural Science Foundation of China (22325304, 22221003 and 22033007), and the Fundamental Research Funds for Central Universities (WK2060000017). We acknowledge the Supercomputing Center of USTC, Hefei Advanced Computing Center, Beijing PARATERA Tech CO., Ltd for providing high-performance computing service. We also thank Dr. Wei Hu, Dr. Xinming Qin, Dr. Mouyi Weng, and Junfeng Qiao for the very helpful discussion.

## Author contributions

B.J. designed the project, and Y.Z. and B.J. discussed the neural network architecture. Y.Z. wrote the code and performed all calculations. Y.Z. and B.J. wrote the manuscript.

## Competing interests

The authors declare no competing interests.
