## [Peer Review File · Nature Communications]

REVIEWER COMMENTS

Reviewer #1 (Remarks to the Author):

The submitted manuscript by Zhang and Jiang proposes a new machine learning interatomic potential (MLIP) that takes into account changes in atomistic systems caused by external fields. Given the fact that most MLIPs are developed for close systems and do not consider the effects of external fields, I believe this work is certainly of interest to many in the field and is well-aligned with the journal. In addition, the machine learning scheme proposed by the authors fulfills the rotational equivariance with respect to the field direction. The authors have provided sufficient proof of the MLIP's capabilities by testing it for systems including the NMA molecule and water. As a conclusion, I recommend publication and only have a few minor comments (please see below).

Minor comments:

- 1) I believe the discussion of the shortcomings of empirical force fields in the introduction could be expanded. In particular, the authors could briefly comment on the ways empirical force fields can take polarization into account and when/why they fail, as well as include some references for these statements. Moreover, the authors state that empirical force fields "fall short to describe bond breakage/formation" which is true for most empirical force fields used, but not for all (ReaxFF for example). Therefore, this statement could be better phrased to avoid generalizations.
- 2) The authors should comment on the computational cost of FIREANN when compared to others machine learning models and especially, when compared to the framework that is based upon, REANN. This would provide an estimate of which computational resources are needed when using a MLIP that takes into account changes from external fields. Additionally, some comments on the changes in computational cost for the training process could be made, mentioning what are the differences in training expense if it is performed with only atomic forces, vs. the most complete scenario (energy, forces, dipole and polarizability tensor).
- 3) Although the authors argue that constructing a model based on atomic forces only could address the multiple-value issue of polarization, I think they should elaborate on the potential shortcomings of doing so for the prediction of some thermodynamic properties, given that the total energy would only be retrieved to an unknown constant.

Reviewer #2 (Remarks to the Author):

The authors introduce an explicit dependence on homogeneous external vector fields into the recursive embedded atom neural network (REANN) machine learning (ML) potential. This makes it possible to describe different response properties as well as interactions of chemical systems with external electric fields with a single model. While the approach is promising and the accompanying experiments are solid, I am concerned that the research presented in the paper is not novel enough to be published in Nature Communications.

The incorporation of external fields into ML models and prediction of response properties has been introduced previously, e.g. in Refs. 54 (for Kernel methods) and 55 (for message passing networks). In general, any model that uses (internally) equivariant feature representations can be easily coupled with an external vector field. This is especially true with the growing prevalence of equivariant ML models and accompanying theoretical tools. Using response formalism to derive different equivariant molecular properties in a consistent way has been explored before as well. In Ref. 55, for example, a single energy model is used to predict forces, dipole moments, polarizabilities and nuclear shielding tensors. The same holds true for the use of such models to simulate vibrational infrared and Raman spectra (including nuclear quantum effects, see e.g. Ref. 55).

The ability to consistently model dipole moments in periodic systems is not very surprising. The ML method essentially implements a data driven decomposition of the global dipole moment vector into local atomic dipoles (see Eq. 4). Using forces computed at different field strengths provides enough indirect information on the charge distribution in the molecule (see e.g. the generalized atomic polar tensor charges in [1]) to train these dipoles. Conceptually similar localization approaches are routinely used to compute infrared spectra of periodic systems. A prominent example are (molecular) dipole moments derived from Wannier localized orbitals, which were also used to compute the electronic structure infrared spectrum shown in Fig. 8 (see SI of Ref. 67). These concepts have also been used in the context of ML methods, such as in [2], which combines a approach based on Wannier localization with a ML driven local dipole decomposition to simulate the infrared spectrum of liquid water.

It should also be mentioned that two of the discussed future developments of of the presented model have already been realized in Ref. 55. Specifically, the inclusion of magnetic fields (nuclear shielding tensors for predicting NMR chemical shifts) and extension to inhomogeneous fields (in the form of solvent environments).

In summary, method development is minimal, field-dependent models and response property prediction have been explored before, as have conceptually similar ML-based dipole decomposition schemes for periodic systems. As a consequence, I cannot recommend the manuscript for publication in Nature Communications.

Comments on the manuscript in general:

Eq. 4 and Fig. 1 are missing the appropriate signs for the response properties. Both forces and dipole moments are the negative derivative of the energy w.r.t. positions and field. Since the polarizability

tensor is defined as the derivative of the dipole moment w.r.t. field, a negative sign is missing there as well.

The way dipole moments were obtained in FIREANN-wD should be explained in more detail. Similarly, more information should be provided for the shifting procedure used to correct the electronic structure dipole moments in the liquid water system.

There is a typing error in line 267, probably the model FIREANN-wD is meant.

[1] Cioslowski, J. (1989). General and unique partitioning of molecular electronic properties into atomic contributions. *Physical review letters*, 62(13), 1469.

[2] Zhang, L., Chen, M., Wu, X., Wang, H., Weinan, E., & Car, R. (2020). Deep neural network for the dielectric response of insulators. *Physical Review B*, 102(4), 041121.

Reviewer #3 (Remarks to the Author):

The paper's central topic is the development of machine learning models for predicting the response of atomic systems under external fields. It is well written and the results are adequately discussed. Pending the addressing of the following comments, I recommend this paper for publication.

Report

Personally, I find the use of Gaussian-type orbitals (GTO) as equivariant mapping for machine learning models interesting. I am aware that the authors have published several papers on this topic, and the current draft extends their previous works by considering an external field. As shown in the results, the application of external-field machine learning models could be helpful for the simulation of spectra.

However, I believe the explanation of the GTO feature map is too condensed for readers who are not familiar with the concept. For instance, in Figure 1, which illustrates the model, it appears that we obtain a density for each atomic environment affected by the other atoms in the molecule. However, according to Eq. 1, it seems that this value is a scalar rather than a feature vector.

Questions and Comments:

- Given that the current model is not the first ML algorithm to incorporate external-field interactions, I think a comparison with FieldSchNet could be beneficial. Additionally, an explanation of both architectures' differences may be relevant.

- Is there a significant difference between the "GTO" feature encoding and standard E(3)-equivariant Graph-NN?
- The paper lacks information about the training for each experiment. For example, the values of λ_V , λ_F , λ_α , and λ_μ are not reported. (Include the learning curves)
- How much does the angular projection of the external field affect the computation of the observables? Does one have to set L to a large number for convergence?
- External field quantum chemistry calculations are expensive and prone to possible systematic errors. Is it fair to assume that one can "warm up" the presented model with external-field-free data and fine-tune it with data that does contain information about the external field?
- Is the discrepancy with experimental spectra in Figure 4 still present even with TRPMD, and would it be improved with more data or is it simply a limitation of the model's learning capacity?
- (Out of curiosity) How well does this model generalize for high-intensity external fields?

Missing citations,

- J. Chem. Theory Comput. 2022, 18, 9, 5492–5501
- Molecules in Electromagnetic Fields: From Ultracold Physics to Controlled Chemistry, Roman V. Krems • Manipulation of molecules with electromagnetic fields, Molecular Physics 111 (12-13), 1648-1682
- J. Phys. Chem. Lett. 2023, 14, 15, 3691–3697
- J. Chem. Theory Comput. 2023, 19, 3, 705–712
- Nat Commun 13, 2453 (2022).

Referee #2 does raise important points regarding the FieldSchNet model (Ref. [55]). As I mentioned in my report, a clear and well-explained comparison between the proposed FieldSchNet model must be included in the revised version, combined with the statement raised by Ref. #2 regarding equivariant ML models for external fields,

This revised version should also present a numerical comparison between both models (and if possible with a kernel method one too) for one of their chemical systems. These results if positive could strengthen the case of their work. The FieldSchNet model is open source so there shouldn't be a limitation to carry those experiments.

We would like to thank all reviewers for their valuable comments. We have carefully addressed all their concerns and made appropriate changes in the revised manuscript. We have provided additional numerical results on the comparison with previous models, as suggested by the referees, and clarify these issues that confused the referees. The point-to-point responses are listed below in blue and the changes in the revised manuscript are given in *Italic*.

Reviewer #1 (Remarks to the Author):

The submitted manuscript by Zhang and Jiang proposes a new machine learning interatomic potential (MLIP) that takes into account changes in atomistic systems caused by external fields. Given the fact that most MLIPs are developed for close systems and do not consider the effects of external fields, I believe this work is certainly of interest to many in the field and is well-aligned with the journal. In addition, the machine learning scheme proposed by the authors fulfills the rotational equivariance with respect to the field direction. The authors have provided sufficient proof of the MLIP's capabilities by testing it for systems including the NMA molecule and water. As a conclusion, I recommend publication and only have a few minor comments (please see below).

Response: We thank this reviewer very much for the recommendation and we have carefully addressed these concerns and properly revised the manuscript.

Minor comments:

1) I believe the discussion of the shortcomings of empirical force fields in the introduction could be expanded. In particular, the authors could briefly comment on the ways empirical force fields can take polarization into account and when/why they fail, as well as include some references for these statements. Moreover, the authors state that empirical force fields "fall short to describe bond breakage/formation" which is true for most empirical force fields used, but not for all (ReaxFF for example). Therefore, this statement could be better phrased to avoid generalizations.

Response: We appreciate the reviewer for these good suggestions and expand the

discussion of the shortcomings of empirical force fields in the introduction. Specifically, we have revised the statements in line 53, “*Although empirical force fields can be instead highly efficient^{21, 22}, their accuracy is limited by empirical functions and approximate expressions for the interaction Hamiltonian. For example, the commonly-used dipole-field approximation truncates the perturbation of the system by an electric field to the first order (i.e. only the interaction with the permanent dipole is included) and omits higher-order interactions associated with polarizability, hyperpolarizability and so on. Moreover, except these reactive force fields²³⁻²⁵, most of them fall short to describe bond breakage/formation.*”

2) The authors should comment on the computational cost of FIREANN when compared to others machine learning models and especially, when compared to the framework that is based upon, REANN. This would provide an estimate of which computational resources are needed when using a MLIP that takes into account changes from external fields. Additionally, some comments on the changes in computational cost for the training process could be made, mentioning what are the differences in training expense if it is performed with only atomic forces, vs. the most complete scenario (energy, forces, dipole and polarizability tensor).

Response: This is indeed a helpful suggestion. We really appreciate it. As evident from Equations (1) and (2), the extra calculation in FIREANN, compared to REANN, arises from a single field-induced orbital. This field-induced orbital involves the calculation of the angular part without the need for evaluating the radial function. As a result, this additional evaluation cost is almost negligible. On the other hand, it is apparent that training in the most complete scenario will be more expensive than training forces only, as the former process requires sample-to-sample high-order gradients (due to polarizability), which have not been optimized in the current implementation of FIREANN built on the original PyTorch framework. This issue could be largely alleviated by an improved implementation based on the more recently released functorch module in a new version of PyTorch, which allows efficient computation of sample-to-sample (high-order) gradients.

In response to these comments, we have added some discussion on the computational cost comparison. In line 124, *“The only extra cost for evaluating FIREANN compared to the standard REANN is that of a field-induced orbital, which is almost negligible as evident from Eqs. (1) and (2).”*

In line 395, *“In the current implementation built on the original PyTorch⁷⁹ framework, training a FIREANN model in the most complete scenario (including energy, forces, dipole and polarizability tensor) will take 4 times longer than force only training, as the former process requires sample-to-sample (high-order) gradients. This issue can be largely alleviated by an improved implementation based on the more recently released functorch⁸⁰ module in a new version of PyTorch, which allows efficient computation of sample-to-sample (high-order) gradients.”*

3) Although the authors argue that constructing a model based on atomic forces only could address the multiple-value issue of polarization, I think they should elaborate on the potential shortcomings of doing so for the prediction of some thermodynamic properties, given that the total energy would only be retrieved to an unknown constant. Response: It is true that training atomic forces only will introduce an undetermined field-dependent constant to the total energy in our model. This will lead to certain undetermined constants to absolute thermodynamic properties. However, atomic forces (or any properties' gradients with respect to atomic coordinates) are well represented by our model so that the changes of thermodynamic properties under a given electric field can still be correctly described, which are physically more meaningful in practical calculations. In addition, by learning the polarizability along with forces, one can eliminate the field dependence of the undetermined constant of the total energy. This allows us to compare energies and the changes of thermodynamic properties under different field strengths, as will be discussed later in the comparison with FieldSchNet. In the revised manuscript, we have added the comments in line 251: *“It should be noted that training atomic forces only will introduce an undetermined field-dependent constant to the total energy in our model. This will lead to certain undetermined constants to absolute thermodynamic properties. However, atomic forces (or any*

properties' gradients with respect to atomic coordinates) are well represented by our model so that the changes of thermodynamic properties under a given electric field can still be correctly described, which are physically more meaningful in practical calculations. In addition, by including the polarizability to the loss function during the training process, one can eliminate the field dependence of the undetermined constant to the total energy. This allows us to compare energies and the changes of thermodynamic properties under different field strengths, as will be discussed later in the comparison with FieldSchNet.”

Reviewer #2 (Remarks to the Author):

The authors introduce an explicit dependence on homogeneous external vector fields into the recursive embedded atom neural network (REANN) machine learning (ML) potential. This makes it possible to describe different response properties as well as interactions of chemical systems with external electric fields with a single model. While the approach is promising and the accompanying experiments are solid, I am concerned that the research presented in the paper is not novel enough to be published in Nature Communications.

Response: We appreciate the reviewer for carefully reading our manuscript and providing critical and constructive feedbacks. These comments are valuable in helping us improve our work. However, we respectfully disagree with the reviewer's assessment on the novelty of our work. We feel that this impression to the reviewer may have arisen from our previous unclear description and insufficient evidence in the manuscript. Below, we have separated reviewer's criticisms into several aspects and made point-to-point responses. Accordingly, we have made significant revisions to our manuscript to better emphasize the originality and superiority of our model. Most importantly, we have included a direct comparison between the FIREANN and FieldSchNet models, demonstrating that FieldSchNet and similar approaches lacking high-order field-system interactions will lead to qualitative failures in some cases. We have provided additional discussion on the limitations of directly learning the Wannier center in periodic systems.

1. The incorporation of external fields into ML models and prediction of response properties has been introduced previously, e.g. in Refs. 54 (for Kernel methods) and 55 (for message passing networks). In general, any model that uses (internally) equivariant feature representations can be easily coupled with an external vector field. This is especially true with the growing prevalence of equivariant ML models and accompanying theoretical tools. Using response formalism to derive different equivariant molecular properties in a consistent way has been explored before as well. In Ref. 55, for example, a single energy model is used to predict forces, dipole moments,

polarizabilities and nuclear shielding tensors. The same holds true for the use of such models to simulate vibrational infrared and Raman spectra (including nuclear quantum effects, see e.g. Ref. 55).

Response: Although the multiple-valued issue can also be overcome by force-only training using the models proposed in Refs. 61 and 62, their ways of incorporating the external field are completely different from the present FIREANN model, rendering the absence of important high-order field-system interactions. Specifically, these models consider the response of the system to the external field by adding the dot product of a virtual atomic dipole and the electric field vector (namely, a scalar value) to the standard atomic descriptor. By construction, high-order field-system interactions are missing in their descriptors. In contrast, by introducing a virtual field-dependent atomic orbital in our field-induced embedded atom density descriptor, we capture the response of the electron density to the external field through orbital-orbital interactions (Eq. (2) in this work). In such a way, all interactions between the field and the atomic environment are included in the our FIREANN model. This is a fundamental improvement over all previous models, which can be easily adapted to other equivariant features based ML models without altering their fundamental architectures.

Indeed, Christensen et al. (Ref. 61) have clearly admitted that their kernel-based model cannot predict polarizability and other high-order response properties. Similar deficiencies arise in the FieldSchNet model as well. Although the nonlinear message passing NNs imposed in that model may learn part of high-order interactions, this incompleteness will cause qualitative failures of FieldSchNet in some cases. To show this explicitly, we have applied the FieldSchNet package to give more numerical evidence. It is found that the FIREANN model, capable of providing a complete description of the response to external fields, perfectly captures the nonlinear energy variation of a water molecule as a function of the field strength. In contrast, the FieldSchNet model predicts no dependence of the energy on the field strength at all. This is because the water molecule lies in the yz plane so that all atomic dipoles derived from FieldSchNet always lie in plane, resulting in zero coupling with the electric field applied along the x -direction and a constant energy. In practice, all x -relevant

components in any response quantities (dipole moment, polarizability, etc.) predicted by FieldSchNet are zero.

This phenomenon is not limited to molecules but also applies to periodic systems. To show this, we construct an exemplary dataset consisting of 200 liquid water configurations exposed to an electric field along the x -direction ranging from 0 to 0.6 V/Å. Water molecules in these configurations are aligned in four evenly spaced layers (16 molecules in each) perpendicular to the x axis with the first and third layer equal to the second and fourth layers respectively, as shown in the inset of Fig. 9. To enable the comparison of field-dependent energies, we trained a FIREANN model and a FieldSchNet model respectively with both forces and polarizabilities (to eliminate the field dependence of the undetermined constant of the total energy), and aligned their field-free energies to the same zero point. The difference between FIREANN and FieldSchNet models are amplified in this system, where the RMSEs predicted on the test set by FIREANN and FieldSchNet are 54.5 meV/Å (2.1 a.u.) and 245.4 meV/Å (165.1 a.u.) for forces (polarizability), respectively. Again, the FIREANN model precisely captures the large energy variance up to an applied electric field of ± 2 V/Å, while the FieldSchNet energy remains constant and deviates from the correct DFT result by several eV, as displayed in Fig. 9.

We note that this deficiency will generally appear in any configuration if all atomic dipoles along the applied field direction are zero, which would lead to an unphysical behavior near the corresponding configuration space and inevitable large fitting errors. For example, using the same full dataset of liquid water in this work and training with atomic forces and polarizabilities, the FIREANN and FieldSchNet models yield test RMSEs of 45.5 meV/Å (2.5 au) and 184.7 meV/Å (12.9 au), respectively. The much worse performance of FieldSchNet represents an indicator of its incomplete description of the field-system interaction. It is worth noting that our FIREANN implementation is more efficient than FieldSchNet, with a training time of 2.4 versus 7.6 minutes per epoch, when running on a single A100 GPU with a memory capacity of 80 GB.

In response to the reviewer's comments, we include these additional results and the aforementioned discussion in the new section "**Comparison with previous models**"

of the revised manuscript from lines 318 to 376.

We have also changed the description in line 159 to “*In comparison, FieldSchNet fails to predict the correct field-induced energy dependence in the same condition and give a constant energy as shown in Fig. 2(c). More detailed comparisons between FIREANN and FieldSchNet will be discussed below.*”

We have added the descriptions for the new dataset of liquid water in line 495, “*To illustrate the deficiency of FieldSchNet, a special dataset was collected consisting of 200 liquid water configurations exposed to an electric field along the x-direction ranging from 0 to 0.6 V/Å. Water molecules in these configurations were averagely placed in four evenly spaced layers perpendicular to the x axis (16 molecules in each). In addition, the first (second) layer was made identical to the third (fourth) layer. In this way, the sum of dipole moments in each layer was kept in plane and any interlayer dipole moment cancelled out, leaving a zero x-component of the total dipole moment. In the data sampling, water molecules were centered evenly spaced grids (4×4) in the plane with some random shifts within 0.3 Å and random in-plane orientation. The intramolecular O-H bond lengths and H-O-H angle of each molecule are randomly displaced from their equilibrium values within ~0.1 Å and ~2°.*”

Fig. 9 Field-dependent energy curves of FIREANN and FieldSchNet. DFT and FIREANN energy curves varying with the electric field intensity, compared with

FieldSchNet for liquid water. The inset is the employed structure of liquid water. Note that in order to compare the energy curves generated by the different methods, the energy curves are shifted separately so that the field-free energy becomes the zero point of the energy.

2. The ability to consistently model dipole moments in periodic systems is not very surprising. The ML method essentially implements a data driven decomposition of the global dipole moment vector into local atomic dipoles (see Eq. 4). Using forces computed at different field strengths provides enough indirect information on the charge distribution in the molecule (see e.g. the generalized atomic polar tensor charges in [1]) to train these dipoles. Conceptually similar localization approaches are routinely used to compute infrared spectra of periodic systems. A prominent example are (molecular) dipole moments derived from Wannier localized orbitals, which were also used to compute the electronic structure infrared spectrum shown in Fig. 8 (see SI of Ref. 67). These concepts have also been used in the context of ML methods, such as in [2], which combines a approach based on Wannier localization with a ML driven local dipole decomposition to simulate the infrared spectrum of liquid water.

Response: We are pleased that the reviewer agrees that forces at different field strengths implicitly capture the response of the charge distribution to external fields, enabling us to reconstruct dipole moments. It is important to emphasize that accurate field-system interactions are essential for force-only training, as we have discussed previously. This ensures both physically correct behavior and high prediction accuracy for dipole moment. However, our approach differs conceptually from existing ML models that rely on decomposing the global dipole moment vector into local atomic dipoles [e.g., Ref. 49 and Ref. 72]. Those models use the product of atomic charges and position vectors to represent atomic dipoles without an external field, following the concept of dipole decomposition. In contrast, we utilize the response formalism, which expresses dipole moment through the energy gradient with respect to the field vector. Although

the reviewer appears to be aware of our model's basis in the response formalism, there seems to be confusion regarding these two distinct concepts. We have changed Eq. (4) to a general response formalism in the revised manuscript to avoid this confusion. This equation itself is valid and does not have to be realized in the atomistic representation.

More importantly, it should be noted that learning dipole moment by the sum of atomic dipoles suffers from the multi-valued issue of the dipole moment in periodic systems. Similarly, the same issue should also appear in the directly learning Wannier centers. Although learning the offsets of the Wannier centers relative to the corresponding O atoms can bypass this contradiction [*e.g.* in Ref. 73], finding Wannier centers themselves in more complex systems, *e.g.* with strong non-local features, is not a trivial task, which easily gets stuck in local minima and has severe convergence difficulties [Ref. 74]. Furthermore, unlike FIREANN, the previous ML model for learning Wannier centers is designed for field-free systems only, which does not describe the general response of a system to external fields and the field-dependent potential energy surface.

In response to the reviewer's comments, we have revised the manuscripts in line 237, "*Similarly, learning the offsets of the Wannier centers relative to the corresponding O atoms can overcome this problem⁷³, however, localizing Wannier centers themselves in more complex systems with strong non-local features, is not a trivial task, which easily gets stuck in local minima and has severe convergence difficulties⁷⁴. Furthermore, unlike FIREANN, these previous ML models are designed for field-free systems only, which does not describe the general response of a system to external fields and the field-dependent potential energy surface.*".

In line 225, "*Fig. 6 (a) shows clearly the abrupt discontinuities in the evolution of the x-component of DFT calculated dipole moment along an AIMD trajectory without an electric field. This accidental change in the dipole moment poses challenges for conventional atomistic ML models that learn field-free dipoles, which typically decompose the global dipole moment vector into local atomic dipoles and represent atomic dipoles by the product of atomic charges and position vectors^{49, 50, 52}.*".

3. It should also be mentioned that two of the discussed future developments of the presented model have already been realized in Ref. 55. Specifically, the inclusion of magnetic fields (nuclear shielding tensors for predicting NMR chemical shifts) and extension to inhomogeneous fields (in the form of solvent environments).

Response: Since our approach encompassing a complete description of external field effects, it is justified to discuss further developments of the FIREANN model itself. In this regard, high-order interactions of magnetic fields with the system can be better realized in the FIREANN framework than in FieldSchNet in the future. In addition, in Ref. 62, the inhomogeneous electric field is approximately generated by solvent environments, which acts only onto the molecular center in a QM/MM-like framework. What we plan to establish is a ML model that introduces an external inhomogeneous field interacting with the entire periodic system (*e.g.* generated by the tip-enhanced confined field or electrochemical interface). This has not been done in any previous work yet.

In response to this comment, we revise the discussion on further developments in the revised manuscript in line 402, “*the FIREANN framework can be extended to describe the response of the system to a magnetic field or even to an electromagnetic field by introducing another field vector-dependent virtual function in Eq. (2). This will allow a more complete description of magnetic fields interacting with the system than in Ref. 62.*” and in line 413, “*Note that this adjustment is intended to introduce an external inhomogeneous field interacting with the entire system. This differs from an inhomogeneous electric field approximately generated by solvent environments, which acts only onto the embedding molecular center as described in Ref. 62.*”

4. In summary, method development is minimal, field-dependent models and response property prediction have been explored before, as have conceptually similar ML-based dipole decomposition schemes for periodic systems. As a consequence, I cannot recommend the manuscript for publication in Nature Communications.

Response: We respectfully disagree with the reviewer's assessment and have made revisions to the manuscript to clearly demonstrate that our method development plays

a crucial role in providing a universal and comprehensive description of the response to external fields. Furthermore, it is important to note that our approach is conceptually distinct from previous ML models.

Comments on the manuscript in general:

5. Eq. 4 and Fig. 1 are missing the appropriate signs for the response properties. Both forces and dipole moments are the negative derivative of the energy w.r.t. positions and field. Since the polarizability tensor is defined as the derivative of the dipole moment w.r.t. field, a negative sign is missing there as well.

Response: We appreciate this suggestion. We have added a negative sign to the derivative in the definitions of force, dipole moment, and polarizability.

6. The way dipole moments were obtained in FIREANN-wD should be explained in more detail. Similarly, more information should be provided for the shifting procedure used to correct the electronic structure dipole moments in the liquid water system.

Response: Both FIREANN-wD and FIREANN-wF models yield the dipole moment in terms of the energy gradient with respect to the electric field. However, the FIREANN-wD model was trained solely with dipole moments, while the FIREANN-wF model was trained solely with atomic forces.

In response to the reviewer's comment, we have included an additional explanation in line 285 to clarify the approach for obtaining the dipole moment in both models, *"Note that dipole moments in both two models are obtained by calculating the energy gradient with respect to the electric field."*

We have also provided more information on the shifting procedure used to correct the ab initio dipole moment in line 269 by *"It is also beneficial to compare the dipole moments predicted by FIREANN-wF with DFT data. Since dipole moments should smoothly change as the configuration evolves, it is reasonable to correct any abrupt changes in the dipole moment calculated by DFT along an AIMD trajectory. This correction involves shifting the dipole moment by an integer multiplied by the product of the corresponding lattice vector and electronic charge. By applying this correction,*

one ensures that the adjusted dipole moment remains closest to its value in the previous step, allowing for a continuous variation of dipole moments along the trajectory.”

7. There is a typing error in line 267, probably the model FIREANN-wD is meant.

Response: Corrected.

[1] Cioslowski, J. (1989). General and unique partitioning of molecular electronic properties into atomic contributions. *Physical review letters*, 62(13), 1469.

[2] Zhang, L., Chen, M., Wu, X., Wang, H., Weinan, E., & Car, R. (2020). Deep neural network for the dielectric response of insulators. *Physical Review B*, 102(4), 041121.

Reviewer #3 (Remarks to the Author):

The paper's central topic is the development of machine learning models for predicting the response of atomic systems under external fields. It is well written and the results are adequately discussed. Pending the addressing of the following comments, I recommend this paper for publication.

Response: We thank the reviewer for this positive assessment.

Report

Personally, I find the use of Gaussian-type orbitals (GTO) as equivariant mapping for machine learning models interesting. I am aware that the authors have published several papers on this topic, and the current draft extends their previous works by considering an external field. As shown in the results, the application of external-field machine learning models could be helpful for the simulation of spectra. However, I believe the explanation of the GTO feature map is too condensed for readers who are not familiar with the concept. For instance, in Figure 1, which illustrates the model, it appears that we obtain a density for each atomic environment affected by the other atoms in the molecule. However, according to Eq. 1, it seems that this value is a scalar rather than a feature vector.

Response: This suggestion is well accepted. The reviewer is correct that one would obtain a density feature for each atomic environment affected by the other atoms in the molecule. It is virtually a feature vector consisting of different density values generated by the square of the linear combination of different GTOs. Eq. (2) just shows how to obtain one component of this vector for a given set of contracted GTOs, as explained in Eq. (7) and associated text in the Methods Section.

We have added the subscript n to the density (ρ) and contracted coefficients (d_m) to denote the n th component of the density vector of atom i . To make this clearer, we have revised the Methods section in line 427, “*which are simply evaluated by the square of the linear combination of a set of contracted Gaussian-type orbitals (GTOs) located at neighbor atoms,*

$$\rho_i^n = \sum_{l=0}^L \sum_{l_x, l_y, l_z}^l \frac{l!}{l_x! l_y! l_z!} \left[\sum_{j \neq i}^{N_c} c_j \sum_{m=1}^{N_\phi} d_m^n \phi_{l_x l_y l_z}^m(\vec{\mathbf{r}}_{ij}) \right]^2 \quad (6)$$

where the primitive GTO takes the following form,

$$\phi_{l_x l_y l_z}^m(\vec{\mathbf{r}}_{ij}) = (x_{ij})^{l_x} (y_{ij})^{l_y} (z_{ij})^{l_z} \exp\left[-\alpha_m (r_{ij} - r_m)^2\right] f_c(r_{ij}), \quad (7)$$

and the contraction combines different shapes of primitive GTOs together,

$$\chi^n(\vec{\mathbf{r}}_{ij}) = \sum_{m=1}^{N_\phi} d_m^n \phi_{l_x l_y l_z}^m(\vec{\mathbf{r}}_{ij}), \quad (8)$$

In practical implementation, we reorder the summation over c_j and d_m^n in Eq (6) to obtain better efficiency. It should be noted that an EAD feature vector (\mathbf{p}_i) consists of a number of density values generated from different sets of contracted GTOs. Although GTOs in our model are expanded in Cartesian coordinates, they can also be expressed in terms of spherical harmonics, resembling in spirit those equivariant features based on spherical harmonics^{44, 48, 53}.

We have also added a sentence in the main text in line 104, “Following the procedure of feature construction in Methods, the field-dependent orbital was combined into the GTO to form a field-induced EAD (FI-EAD) vector that comprises various density values determined by different set of contracted coefficients,

$$\rho_i^n = \sum_{l=0}^L \sum_{l_x, l_y, l_z}^{l_x+l_y+l_z=l} \frac{l!}{l_x! l_y! l_z!} \left[\sum_{m=1}^{N_\phi} d_m^n \left(\sum_{j \neq i}^{N_c} c_j \phi_{l_x l_y l_z}^m(\vec{\mathbf{r}}_{ij}) + c_\varepsilon \phi_{l_x l_y l_z}(\vec{\boldsymbol{\varepsilon}}_i) \right) \right]^2 \quad (2)$$

”

Questions and Comments:

- Given that the current model is not the first ML algorithm to incorporate external-field interactions, I think a comparison with FieldSchNet could be beneficial. Additionally, an explanation of both architectures’ differences may be relevant.

Response: We really appreciate this referee for this valuable suggestion, which have been addressed in very detail in our responses to Reviewer #2. Indeed, additional results on this comparison clearly demonstrate the superiority of the current model over FieldSchNet.

- Is there a significant difference between the “GTO” feature encoding and standard E(3)-equivariant Graph-NN?

Response: Gaussian-type orbitals (GTOs) are conceptually similar as the commonly used equivariant features based on spherical harmonics. Indeed, GTOs in our model are expanded in Cartesian coordinates, which can also be expressed in terms of spherical harmonics. The primary difference exists in the architectural framework of the two models. E(3)-equivariant Graph-NN belongs to the equivariant message-passing NN, which passes both invariant and equivariant features. Whereas FIREANN/REANN is essentially an invariant message-passing NN, which only passes invariant information.

- The paper lacks information about the training for each experiment. For example, the values of λ_V , λ_F , λ_α , and λ_μ are not reported. (Include the learning curves)

Response: We have now provided a list of weights for each target in line 465, “*These weights for individual properties in the loss function were dynamically adjusted during the training process. For the NMA molecule, λ_V , λ_F , λ_μ and λ_α decay linearly from 0.1, 50, 10, and 10 to 0.1, 0.5, 0.5, and 0.5 as the learning rate decays. The same set of weights were used for the H₂O molecule, except that there is no force weight included. As for the liquid water, only atomic forces were trained and weighting was unnecessary.*”

- How much does the angular projection of the external field affect the computation of the observables? Does one have to set L to a large number for convergence?

Response: In our experiences, the angular projection converges very rapidly and in most cases $L=2$ is sufficient to obtain a very accurate model.

- External field quantum chemistry calculations are expensive and prone to possible systematic errors. Is it fair to assume that one can “warm up” the presented model with external-field-free data and fine-tune it with data that does contain information about

the external field?

Response: This is a great suggestion, especially when ab initio data are computationally expensive. In our research, we found that the quantum chemistry calculations under external fields performed by CP2K for systems such as liquid water was not significantly more expensive compared to the field-free calculations. Additionally, it is crucial to highlight that simulations conducted under external fields enable exploration of a much wider dynamically relevant configurational space, especially in the case of liquid water. These configurations are typically challenging to be sampled in field-free calculations. We will explore the suggested strategy more systematically in future work.

- Is the discrepancy with experimental spectra in Figure 4 still present even with TRPMD, and would it be improved with more data or is it simply a limitation of the model's learning capacity?

Response: We do observe the discrepancy with experiment even employing the TRPMD method, which is more likely due to the error of DFT or the approximation in TRPMD rather than the model accuracy. Indeed, Notably, our findings using the FIREANN-wF model successfully reproduce the on-the-fly results obtained at the same DFT setup and through TRPMD simulations, as demonstrated in Fig 8. We have also verified that the predicted spectrum converges well with the dataset size, indicating that additional data would not significantly enhance the results.

- (Out of curiosity) How well does this model generalize for high-intensity external fields?

Response: A correct symmetry adaption or physical constraint can always enhance the generalizability of a ML model. In the present case, the FIREANN model trained with the field strength within the range of 0 to 0.6 V/Å perfectly reproduces the energy and dipole moment of liquid water across an electric field range of -2.0 V/Å to 2.0 V/Å, as depicted in the Fig. 9. This exemplifies the generalizability of this model towards representing high-intensity external fields.

We have added a statement in the revised manuscript in line 360, “*Again, the FIREANN model precisely captures the large energy variance up to an applied electric field of ± 2 V/Å, while the FieldSchNet energy remains constant and deviates from the correct DFT result by several eV, as displayed in Fig. 9. This result also validates the generalizability of the FIREANN model towards representing high-intensity external fields.*”

Missing citations,

- J. Chem. Theory Comput. 2022, 18, 9, 5492–5501
- Molecules in Electromagnetic Fields: From Ultracold Physics to Controlled Chemistry, Roman V. Krems • Manipulation of molecules with electromagnetic fields, Molecular Physics 111 (12-13), 1648-1682
- J. Phys. Chem. Lett. 2023, 14, 15, 3691–3697
- J. Chem. Theory Comput. 2023, 19, 3, 705–712
- Nat Commun 13, 2453 (2022).

Response: We thank the reviewers for suggesting these citations. We have incorporated all recommended references except the third one [J. Phys. Chem. Lett. 2023, 14, 15, 3691]. While we acknowledge its contribution in utilizing vibrational spectroscopy to identify materials with well-defined stoichiometry and separate genuine vibrational features from morphological and defect-induced signals, we do not think it is relevant to the subject of our work.

In addition, we have discussed the fourth reference in the line 233, “*Schienbein also recognized the multiple-valued problem of the dipole moment and proposed to learn atomic polar tensor which is the spatial derivative of dipole instead of learning the dipole itself. These smooth spatial derivatives can be transformed into time-derivatives of dipole in molecular dynamics to calculate autocorrelation functions, ultimately yielding the IR spectrum⁶⁰.*”

REVIEWERS' COMMENTS

Reviewer #1 (Remarks to the Author):

The authors have addressed all my concerns and comments in their revision, therefore I recommend publication in Nature Communications.

Reviewer #3 (Remarks to the Author):

The authors addressed all my comments and suggestions and given this new version, the draft could be publishable.

Reviewer #1 (Remarks to the Author):

The authors have addressed all my concerns and comments in their revision, therefore I recommend publication in Nature Communications.

Response: We thank this reviewer very much for the recommendation.

Reviewer #3 (Remarks to the Author):

The authors addressed all my comments and suggestions and given this new version, the draft could be publishable.

Response: We thank this reviewer very much for the recommendation.